# 'I cannot be what I don't see': an evaluation of Academic Intersectionality Mentoring in medical schools (AIMMS Mentoring)

Mirembe Woodrow[1], Elizabeth Benedikz[2], Louise D Bryant[3], Jane Illés[4], Parbir Jagpal[5], Hannah Maria Jennings[6], Eleanor Monks[7], Vrinda Nayak[8], Musarrat Maisha Reza[8], Sikha Saha[9], Meena Upadhyaya[10], Kate Williams[11], John P Winpenny[4], Reza Zamani[8], Nisreen A Alwan[1,12,13]*

1 School of Primary Care, Population Sciences and Medical Education, Faculty of Medicine, University of Southampton, Southampton, United Kingdom, 2 Academy of Medical Sciences, London, United Kingdom, 3 Faculty of Medicine and Health, University of Leeds, Leeds, United Kingdom, 4 School of Medicine, University of St Andrews, St Andrews, United Kingdom, 5 School of Pharmacy, College of Medical and Dental Sciences, University of Birmingham, Birmingham, United Kingdom, 6 Hull York Medical School, University of York, York, United Kingdom, 7 School of Healthcare Enterprise and Innovation, Faculty of Medicine, University of Southampton, Southampton, United Kingdom, 8 University of Exeter Medical School, Faculty of Health and Life Sciences, University of Exeter, Exeter, United Kingdom, 9 Institute of Cardiovascular and Metabolic Medicine, School of Medicine, University of Leeds, Leeds, United Kingdom, 10 Division of Cancer and Genetics, Cardiff University, Cardiff, United Kingdom, 11 Leicester Medical School, University of Leicester, Leicester, United Kingdom, 12 NIHR Southampton Biomedical Research Centre, University of Southampton and University Hospital Southampton NHS Foundation Trust, Southampton, United Kingdom, 13 NIHR Applied Research Collaboration Wessex, Southampton, United Kingdom

* N.A.Alwan@soton.ac.uk; nisreen_alwan@outlook.com

## Abstract

The Academic Intersectionality Mentoring in Medical Schools (AIMMS Mentoring) scheme aims to support the career progression of women from ethnic minority backgrounds working in academic medicine and health sciences who are under-represented in senior roles of academia in the UK. Two questionnaires (baseline and 6 months) captured information about AIMMS Mentoring participants and practical aspects of the scheme. Participants were asked about their experience of and satisfaction with the scheme, whether it matched their expectations and what they felt were the scheme's rewards and challenges. Questions were also asked about the organisation of the scheme and how it could be improved. The productivity of mentoring relationships was explored, including personal and professional development, and whether participants felt it important that mentoring takes place between people with similar characteristics. Sixteen pairs took part in mentoring, with ten mentees and four mentors completing evaluation questionnaires at 6m follow-up. Responses indicated that the scheme was helpful. All mentor and mentee responses reported personal and professional development. Mentee responses in particular reported gaining insight into career development, and both groups felt they had gained understanding about institutional ways of working.

**Data availability statement:** The dataset includes the individual responses to the two surveys used for this evaluation. The data owner is the University of Southampton. The study data are available on request provided ethics committee approval for sharing the anonymized data is granted. To request access conditional on approval, please email "rgoinfo@soton.ac.uk" the University of Southampton's Research Integrity and Governance team.

**Funding:** The author(s) received no specific funding for this work.

**Competing interests:** The authors have declared that no competing interests exist.

Participants rated the scheme positively and indicated they would recommend it to others. Being in mentoring relationships with women from similar backgrounds was ascribed value, as was mentor partners being empathetic. The evaluation revealed ways in which the scheme could be improved. Women from ethnic minorities working in academic medicine and health sciences can face structural barriers into leadership. This formative and summative evaluation of AIMMS Mentoring showed that mentoring and peer support in this group is valuable and can assist with personal, professional and career development. The scheme is an example of positive action and a model national activity aimed at achieving equity of opportunity in academic medicine and health sciences.

## Introduction

Approximately 18% of the population of England and Wales belong to a black, Asian, mixed or other ethnic group [1]. If this demography was reflected amongst UK academic professors, we would expect approximately 9% of professors to be female and belonging to ethnic minorities groups. Latest data demonstrate that the proportion of professors who are women from ethnic minorities falls far short of this, with about two-thirds (63.6%) of professors white males, about one quarter (26.5%) white female, 7.2% males from ethnic minority backgrounds and only 2.7% females from ethnic minority backgrounds [2].

The concept of intersectionality acknowledges that no individual has a single-dimension identity and that social constructs shaping identity dimensions can lead to compound disadvantage [3]. Intersectionality critical theory conceptualises knowledge as contextual, relational, and reflective of political and economic power [4]. For example, the experiences of a lesbian black woman are vastly different to those of a heterosexual white woman, even though they share one dimension of their identity (gender). Multiple other identity dimensions may shape life experiences, such as disability, caring responsibilities, language, or migration status, and may overlap in a way that exacerbates the experiences of disadvantage and hardship. Addressing inequities and structural discrimination through the lens of intersectionality is not easy, though much needed. Analyses of research funding programmes show that disparities span across more than one identity dimension [5]. For example, over the course of seven decades, the prestigious Lasker biomedical research award was given to only one non-white woman out of 397 [6]. Intersectional inequalities can span across pay, recognition and promotion, and the answer is not to prioritise one identity dimension over the other as some initiatives do (e.g., gender over ethnicity [5]), but to adopt an intersectional approach to addressing inequities entrenched in the sector.

The most significant initiative developed to redress gender imbalance in UK higher education, Athena Swan [7], has recently increased its focus on intersectionality; however clear impact cases are still rare. Still relatively new, Advance HE's Race Equality Charter [8] and the BMA's Racial Harassment Charter for Medical

Schools [9], which provide frameworks for institutions' action against discrimination, have not yet enabled large-scale change in the race profile of staff in Higher Education Institutions (HEI) or academic medicine.

There is evidence that having a same-gender role model is influential for early-career women in some medical specialties [10]. Mentorship is an activity in which professionals engage to help develop the next generation of professionals in their field [11] and a popular initiative widely regarded as being a useful way to try to address inequalities in the workplace [12]. However, there is limited evidence around how mentoring can help achieve equity in the higher education sector or on whether matching mentor/mentees based on intersectional backgrounds is beneficial, and, if so, how. Evidence of mentoring effectiveness in reducing gender inequalities is largely lacking and evaluation of mentoring schemes needs to be more robust [12]. This paper describes the formation, design and evaluation of the Academic Intersectionality Mentoring in Medical Schools scheme (AIMMS Mentoring).

## The AIMMS Mentoring scheme

AIMMS Mentoring was established in 2020 following an informal process of identifying need through conversations around intersectionality among women from ethnic minorities in academic medicine at the University of Southampton led by the Founding Chair of the scheme (NAA). It was felt that this type of intersectional mentoring could only be accomplished at a national, rather than a single institution, level due to lack of sufficient number of ethnic minority women at senior levels in any one institution. NAA reached out to other UK medical schools to explore interest in forming such an initiative. The founding members were representatives from nine UK medical schools in partnership with the Academy of Medical Sciences. Members included The University of Birmingham, Cardiff University, The University of Exeter, Hull York Medical School, The University of Leeds, The University of Leicester, The University of St Andrews and Swansea University, chaired by NAA at the University of Southampton.

The AIMMS Founding Committee contributed to designing the scheme, its conduct and its evaluation and promoted it through formal and informal networks to all medical schools and health science faculties across the UK. The committee provided advice and steering throughout the establishment process and continues to meet regularly as the scheme evolves and expands. University of Southampton staff administered the scheme.

## AIMMS Mentoring aims and objectives

The aim of AIMMS Mentoring is to support the career development of minority ethnic women towards equal and authentic leadership in academic medicine and health sciences. It is a unique scheme in the UK and the first national medical schools' scheme to connect women from ethnic minority backgrounds employed in clinical and non-clinical career pathways. Its objectives include:

• connecting women from ethnic minority backgrounds working in academic medicine and health sciences

• providing participants with a chance to act as mentees and/or mentors and peer support one another

• Increasing visibility of, and access to, role models

• facilitating participants' progression to leadership levels

• becoming an exemplar national activity for participating institutions towards achieving equality of opportunity and meeting the targets of the Athena Swan and Race Equality Charters

The scheme is open to anyone employed in academic medicine or health sciences and who identifies as a woman from a black, Asian or any ethnic minority background. Due to capacity and governance issues, the scheme is currently not available to undergraduate or postgraduate students.

A website about the scheme was built, hosted by the University of Southampton. Members of the committee used university communications to recruit mentors and mentees and promote the scheme widely. They also highlighted it in presentations at relevant conferences and webinars. Interested women contacted the scheme through a dedicated email address.

An evaluation was built into the design of the scheme from its initiation. Members of the AIMMS Founding Committee designed the evaluation using summative and formative principles.

## Scheme procedure

When an expression of interest in the scheme was received from an eligible woman, they were sent a welcome email including detailed information about the scheme, how the matching process worked and the scheme's privacy notice. They were then asked to register their details online using Microsoft Forms, asked to provide written consent to participate in the scheme and to confirm that they would abide by the scheme's code of conduct and confidentiality. Women could choose to be mentors, mentees or both.

Members were also asked to provide information about their career to date and what they could offer as a mentor or, sought from participation as a mentee. Mentors' details were published online and mentees used the list to identify their top three mentor choices. If top preferences were unavailable due to capacity, efforts were made to find an appropriate match. In these situations, or where mentees had not expressed a preference, matches were proposed based on relative seniority and research interests (where possible). Both mentor and mentee were provided with the opportunity to decline the match if they did not think it was suitable. The duration of the matching process ranged from a few days to several weeks, depending on mentors' capacity and participants' speed of response.

Mentors and mentees were given initial guidance about agreeing mentoring arrangements that were acceptable to them both for example frequency of meetings, expectations about what would be discussed, meeting outputs, and were referred to established Academy of Medical Sciences (AMS) guidance [13]. The AMS guidance included training material, advice on adopting different models of mentoring, template mentoring agreements and other tips for productive mentoring. The scheme administrators explained that the first meeting should be on a trial basis so that women did not feel under pressure to pursue a match where they did not feel comfortable. As mentors and mentees were from differing universities, mentoring meetings took place virtually. Mentoring matches were initially for a period of six months, but pairs were welcome to continue mentoring beyond the six month period by mutual agreement.

## Methods

### Ethics approval

Ethics approval for this evaluation was granted by the Ethics Committee of University of Southampton Faculty of Medicine (65315.A1).

### Recruitment and data collection

When women who were interested in registering for the scheme were first emailed details about AIMMS Mentoring, they were informed about the evaluation at the same time. They received a participant information sheet and it was made clear that participation in the evaluation was entirely voluntary. Written consent to participate was sought. Recruitment to the evaluation took place on a rolling basis between 11 February 2022 and 10 March 2023.

Baseline questionnaires were designed for mentors and mentees to complete before mentoring commenced. Six months after the start of mentoring, mentors and mentees were asked to complete a second questionnaire (see Supporting information). Participants were not asked to identify themselves in either questionnaire. The questionnaires requested descriptive data about the participants (job title, ethnicity, age) but answering these questions was optional. Information was also

requested about how participants felt about their mentor/mentee pairing, and the number of mentoring sessions that had taken place. Questions asked about participants' experience of, and satisfaction with, the scheme, whether it matched their expectations, and what they felt were the scheme's rewards and challenges. Questions also asked about the organisation of the scheme, its process and structure, how accessible it was and how it could be improved. We also explored the productiveness and usefulness of the mentoring relationship, including personal and professional development outcomes, and whether participants felt it was important that mentoring takes place between people with similar characteristics.

## Scheme participants

At the time of writing, 88 women had expressed an interest from across 22 UK universities since the scheme was established. Thirty-three went on to register with the scheme, and of those 28 have so far participated in mentoring.

## Analysis

The questionnaires were issued using MS Forms. Descriptive analysis was conducted using MS Excel. This included generating statistics and grouping together themes/statements (where there was free text) from the responses to each question. Free text responses provided the source of quotations selected to illustrate findings.

## Results

By July 2023, 16 mentor/mentee pairs had begun mentoring. Some mentors had capacity to take on more than one mentee at a time. Two mentor pairs were unsuccessful, with mentoring abandoned at an early stage. Mentors who were also AIMMS Mentoring committee members were excluded from taking part in the evaluation. In total, 10 mentee and 10 mentor responses were completed for the baseline assessment, and 9 mentee and 4 mentor responses completed for the follow-up assessment at six months.

Responses showed that there were a mix of roles in terms of seniority, with more senior roles overall being reported in mentor responses. This was the case with age too: mentor responses indicated older age categories than mentees overall. Ethnicity was reported as free text, and responses included Asian, Asian British, Pakistani, British Pakistani, Indian, British Indian, Arab, black African, black Caribbean and white, Latino/European, Chinese, Malaysian Chinese, and Chinese/Vietnamese.

## Baseline questionnaire

Before mentoring started participants were asked **whether having a successful career in academic medicine had been more challenging because they were a woman from an ethnic minority**. Opinions were divided: mentee responses showed an equal split between those who agreed and who disagreed, whereas in mentor responses, seven agreed and three disagreed. Free text comments revealed some themes (Table 1). Illustrative examples of free text comments are provided in Box 1.

> ## Box 1
>
> **Free text comments in response to the question "Do you feel that having a successful career in academic medicine has been more challenging because you are a woman from an ethnic minority?"**
>
> "I cannot be what I don't see" (Mentee response)
>
> "I found myself trying to fit in by either not mentioning a homelife and by trying to be more 'English'" (Mentor response)

**Table 1. Do you feel that having a successful career in academic medicine has been more challenging because you are a woman from an ethnic minority.**

| Yes<br>n = 12 (5 mentee, 7 mentor) | No<br>n = 8 (5 mentee, 3 mentor) |
|---|---|
| ***Themes revealed in free text explanatory comments*** | |
| Differing cultural norms and values had an impact | Challenges participants had experienced were due to other reasons, e.g., research politics, poor early career researcher policies |
| Unconscious bias was at play | Participants had not experienced any challenges due to their identity as a woman from an ethnic minority yet, but anticipated it was possible in the future |
| Participants experienced higher levels of administrative scrutiny, and targeting in redundancy programmes | Participants' research topics were related to ethnic minorities so this gave them an advantage |
| Participants experienced heavier workloads than peers, with less support and less time for career development | Opportunities were available and needed to be 'grabbed' |
| Participants experienced being talked down to, or not taken seriously | Ethnicity bias was worse in their country of origin |
| Participants purposefully did not discuss home/family life at work to demonstrate commitment to professional life | Participants felt their status as a woman was potentially the difficulty, rather than being from an ethnic minority |
| Participants experienced a feeling of isolation or not fitting in, and not being represented or of not being visible | |
| Participants experienced personal and institutional barriers to career progression | |

Before starting mentoring, participants were asked **what they would like to get out of the scheme** (see Table 2).

## Follow-up questionnaire

Nine mentee responses and four mentor responses were completed for the follow-up questionnaire at six months. Mentor follow-up responses showed broadly more senior roles and older age than mentee responses on average. Reported ethnicity included British Asian, British Pakistani, Southeast Asian, Arab, Black African, Latino and Malaysian Chinese.

Participants were asked to **rate the scheme** from 1 to 5 stars. The mean rating in mentee responses was 4.6 stars, whilst mentor responses awarded a mean of 3.8 stars. During the six months, mentoring pairs had met between once and

**Table 2. What do you want to get out of the scheme.**

| *What do you want to get out of the scheme?* | Mentees<br>n = 10 | Mentors<br>n = 10 |
|---|---|---|
| Professional development, e.g., research skills, grant applications | 7 | 4 |
| Personal development, e.g., confidence, leadership | 6 | 6 |
| To develop contacts and networks | 4 | 5 |
| To enhance my CV and/or portfolio | 3 | 2 |
| To receive practical advice about how to progress my career (mentees only) | 9 | N/A |
| To support women from ethnic minorities in academic medicine | 6 | 10 |
| General support (mentees only) | 4 | N/A |
| To increase my understanding about the difficulties facing women from ethnic minorities in academic medicine (mentors only) | N/A | 7 |
| Other | 1<br>Free text:<br>*"How to navigate my career as a woman of ethnic minority"* | 0 |

5 times (mean 2.7 times). Six out of nine pairs were still mentoring after six months. Mentee and mentor responses all agreed or strongly agreed that they had a good relationship with their mentoring partner.

Respondents were asked to rate the helpfulness of the support available from the scheme to undertake mentoring. Mentee responses showed eight of nine agreed or agreed strongly that the support was helpful. In mentor responses, one agreed, two were neutral and one disagreed. Similarly, only 1 of 4 mentor responses agreed they felt supported by others (e.g., their department or colleagues) to take part in the scheme. More mentee responses agreed they felt supported (5 of 9 agreed or strongly agreed they felt supported).

Participants were asked **how the scheme could be improved** and provided several suggestions (Fig 1).

All nine mentee responses agreed that their experience of the mentoring scheme matched their original expectations, and three out of four mentor responses also agreed. Illustrative examples of free text comments are provided in Box 2.

---

Box 2

**Free text comments in response to the question "Did your experience of the scheme match what you expected?"**

"… I think having the opportunity to connect with women who can relate (even just a little) to personal and professional experience has been really good for me." (Mentee response)

"I wanted to help further my mentees career and support queries. I hope I was able to encourage and take her to the next step." (Mentor response)

---

Having a wider range of mentors from different medical and health science backgrounds to enable improved matching of professional interests

More consistent guidance about what could be discussed during mentoring and what to expect

Improved training on mentoring

Providing in-person networking events and more availability of peer workshop sessions

Structured feedback about mentors from mentees that can be used in mentors' own appraisals and professional development

**Fig 1. How could the scheme be improved.**

**Table 3. Mentoring has helped me develop personally/professionally in the following areas.**

| | Mentees n=9 | Mentors n=4 |
|---|---|---|
| *Mentoring has helped me develop personally in the following areas* | | |
| Interpersonal/communication skills | 1 | 0 |
| Listening skills (mentors only) | N/A | 2 |
| Self confidence/validation | 4 | 0 |
| Leadership | 1 | 1 |
| Mentoring skills (mentors only) | N/A | 3 |
| Goal setting | 4 | 1 |
| Collaborative working | 0 | 0 |
| Dealing with conflict | 0 | 0 |
| Negotiating | 2 | 0 |
| Understanding of other perspectives | 1 | 3 |
| Balancing competing needs | 3 | 0 |
| Achieving a good work-life balance | 3 | 0 |
| Developing independence (mentees only) | 1 | N/A |
| Opportunity for self reflection | 4 | 0 |
| I have not developed personally | 0 | 0 |
| Other | 0 | 0 |
| *Mentoring has helped me develop professionally in the following areas* | | |
| Expansion of professional networks | 2 | 1 |
| Academic responsibilities, knowledge and skills | 0 | 0 |
| Management and development skills | 0 | 1 |
| Increased productivity (mentees only) | 1 | N/A |
| Performance at job interviews (mentees only) | 0 | N/A |
| CV development | 2 | 0 |
| Insight into career development opportunities, promotion criteria, methods | 4 | 0 |
| Insight into institutional ways of working | 5 | 3 |
| Insight into the nature of academic life as a woman from an ethnic minority | 3 | 0 |
| I have not developed professionally | 0 | 0 |
| Other | 0 | 0 |

Mentees and mentors were asked **what they found most rewarding and most challenging from mentoring** and responded in free text with a variety of themes (Fig 2 and Fig 3).

Table 3 describes **how mentors and mentees had developed personally and professionally through mentoring**. All responses indicated development in some way. Personal development in self-confidence, goal setting and reflection were more common in mentee responses, and listening, mentoring and understanding perspectives were more common in mentor responses. Professionally, mentee responses indicated gaining insight into career development and both mentor and mentee responses reported gaining insight into institutional working. Illustrative examples of free text comments are provided in Box 3.

In free text comments, one mentee response reported that mentoring had helped with a successful promotion. Another mentioned that goal setting through mentoring had enabled more focus.

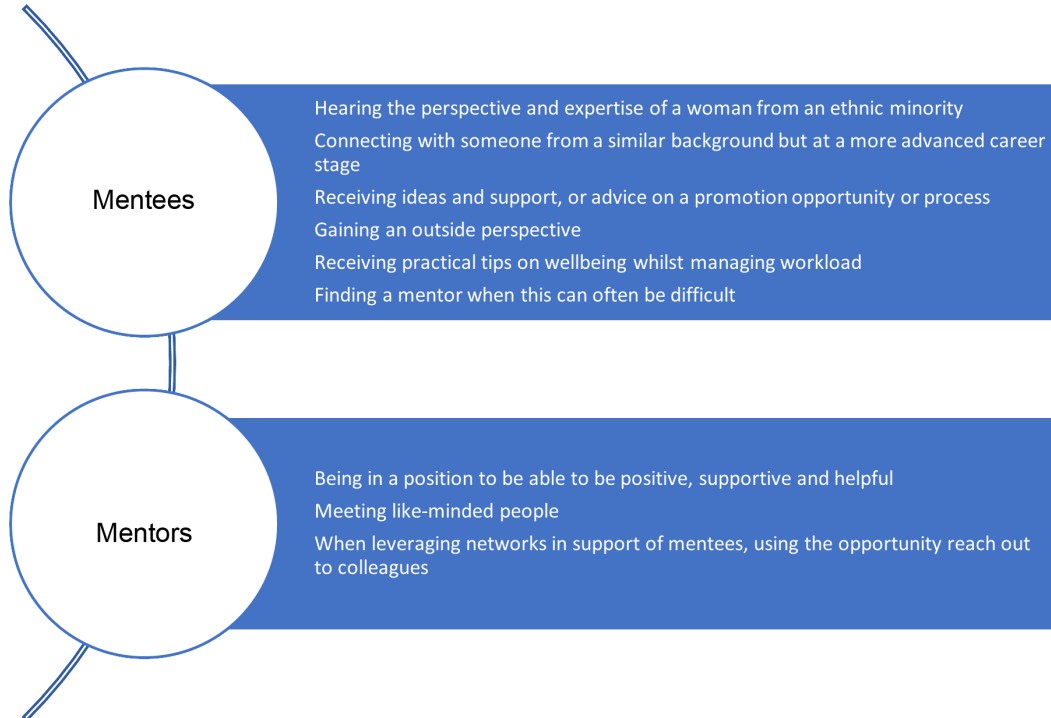

**Fig 2. What did you find most rewarding about mentoring.**

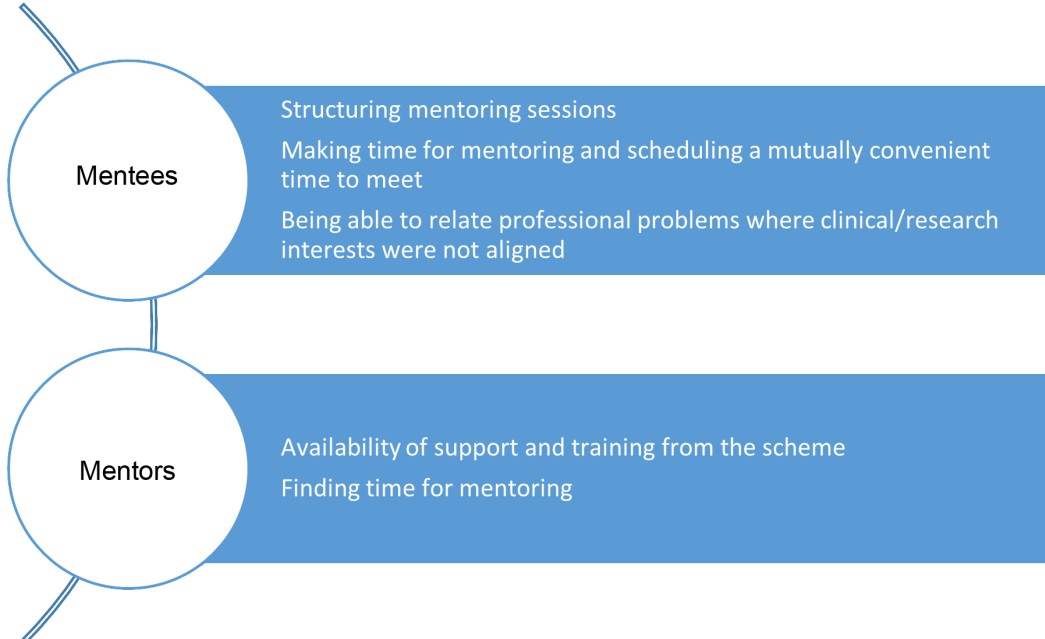

**Fig 3. What did you find most challenging about mentoring.**

> **Box 3**
>
> **Free text comments in response to the question "Mentoring has helped me develop personally in the following areas"**
>
> "I have achieved formal recognition and promotion for the work I have done which is a great boost to my morale and career." Mentee respondent
>
> "Having the mentor meetings helped me set goals, so that every time we met, we agreed that I would have completed the goal, and actually, these little goal-setting steps and then meeting up, really helped me stay focused." Mentee respondent

All responses at six months follow up indicated that mentoring had been taken part in previously and some had taken part as both a mentee and a mentor. When asked, having participated in this mentoring scheme how **confident they were that it would positively impact on them or their career**, eight of nine mentee responses gave a score of 7 or above (where 10 is extremely confident), while mentor responses generally gave lower scores of 5–8.

Having participated in the scheme, mentees and mentors were asked **how useful it is that mentoring takes place between people with similar characteristics**. Eight of nine mentee responses rated this a score of 7 or above (with four rating it 10 'very useful indeed') while mentor responses gave scores ranging between 5 and 10. Illustrative examples of free text comments are provided in Box 4.

> **Box 4**
>
> **Free text comments in response to the question "How useful is it that mentoring takes place between people with similar characteristic?"**
>
> "It has been so beneficial to me to speak with someone who can relate to my experience. In the past, I have been matched with a mentor (middle-aged, white, male, business-oriented rather than research) who I did not get on with as it felt like I had nothing in common with him. I think it's been especially beneficial to have someone who has had a similar career trajectory/specialised in a similar field as this has facilitated conversations massively." (Mentee response)
>
> "Speaking with my mentor has given me insight into ways that I could progress in my career, also how I can use social media platforms to aid in a research career. It was also helpful to hear about their experience of progressing in academia but in a different institution, it gave me a good insight into how things work in different places." (Mentee response)

At six months, all respondents reported they would recommend the scheme to others. Illustrative examples of free text comments are provided in Box 5.

> **Box 5–Free text comments in response to the question "Would you recommend this scheme to others?"**
>
> "I think this scheme could really help to empower younger women from ethnic minorities in academia" (Mentee response)

## Discussion

We established a national scheme to match academic staff working in medicine and health sciences who identify as women from ethnic minority backgrounds in peer mentoring relationships. Women from a range of ethnicities, roles and age groups took part. Before starting mentoring, participants indicated that they hoped the scheme would help with their personal and professional development, with developing networks and with career progression. When asked whether participants felt that having a successful career in academic medicine had been more challenging due to being a woman from an ethnic minority (the question did not state a comparator), a range of views were expressed. Some women felt their career had been impacted negatively, for example through experiencing discrimination, isolation and lack of representation in the workplace. Others had not experienced negative impacts in the UK resulting from their identity, or at least not so far in their careers, or had experienced worse discrimination in other countries. Some made the point that career opportunities were available to be taken advantage of, or that their identity gave them an advantage in their own research relating to ethnicity.

The evaluation has shown that participants found the scheme helpful and expectations had been met in several ways. Participants felt that taking part would have a positive impact on their career. After six months, all mentors and mentees who completed follow-up reported that they had developed personally and professionally in a number of ways. Mentees in particular reported that they had had opportunities for self-reflection, goal-setting and confidence-building. They had also gained insight into career development and both groups felt they had gained understanding about institutional ways of working. All participants gave the scheme a positive rating and indicated they would recommend it to others. They also reported that they valued being in mentoring relationships with women from similar backgrounds. For these women it was important that mentorship partners have empathy.

The evaluation also identified ways in which the scheme could be improved, most notably by the provision of additional support and mentoring training for mentors and mentees. Tailored training sessions have recently been implemented as part of AIMMS Mentoring and informal feedback shows they are highly valued. A training webinar will also be included in the induction pack of prospective participants, in addition to the online support resources links provided. There was also the suggestion that an expansion of the scheme to include more women from a range of clinical and non-clinical backgrounds would be beneficial. The AIMMS Mentoring Founding Committee is exploring sustainable options to widen it to include other intersectional groups.

Although the evaluation involved small numbers of participants, the objectives of AIMMS Mentoring have broadly been shown to be achieved by this evaluation. The scheme has connected women from ethnic minority backgrounds working in academic medicine and health sciences at different institutions, and has provided them with the opportunity to act as mentors/mentees. Although not the subject of the evaluation the scheme also provides sessions to enable women to support one another as peers. AIMMS Mentoring has introduced role models through its mentors, and enabled personal and professional development. The evaluation showed that in at least one case it facilitated a mentee's progression into a leadership role.

## Limitations of the study

The evaluation was limited by its small numbers and therefore cannot claim to be representative of wider experiences. It was also important not to overburden the participants, so the questionnaires were kept as short as possible. Interviews in addition to the questionnaires may have achieved richer data, but would have placed additional burden on women who already reported being time poor. Resources for this study were very limited, which restricted the design as well the limited the assessment to short-term outcomes. For example, adding a control group who did not take part in mentoring would have been valuable, as would a longer-term cohort study to investigate longer-term impacts.

Future research could include understanding the specific elements of an intersectional approach to mentoring that may help address structural inequity within the higher education context. We recognise that the burden should not be placed

on those disadvantaged by structural barriers to 'fix' the system. Rather this should be a collective effort driven by senior leadership through sustainable policies and strategies.

## Conclusions

Our results underline that women from ethnic minorities face barriers to progression into leadership roles. More resources like AIMMS Mentoring are needed to provide support and address the lack of balanced representation at senior levels. It is worth noting that some participants did not feel adequately supported by their work environment to dedicate time towards participating in this scheme. Support is needed from managers and institutions that enables women to participate in such schemes by raising awareness, providing encouragement and protection of time to take part. Senior leadership commitment to action is needed across gender, race and disability at institutional level [14] as well as a supportive and inclusive workplace culture [15].

The AIMMS Mentoring Founding Committee believe that bringing multiple medical schools together with a common goal was important in itself. The collaborative effort to pursue the establishment of the scheme recognised the national dearth of women from ethnic minorities in leadership positions in UK medical schools and that this needs to change. Student cohorts are becoming more diverse, and it is crucial that senior role models with whom ethnic minority students identify become more common and more visible. Without this, medical education will not progress in the way it needs to.

The scheme can be viewed as an exemplar national activity towards achieving equity of opportunity in academic medicine and health sciences. As a unique scheme in the UK, AIMMS Mentoring is a valuable asset. It can be adapted and expanded to other academic disciplines, within the UK and globally, as well as towards supporting other marginalised intersectional groups.

## Supporting information

**S1 Supporting information.** **Baseline and 6m follow-up questionnaires for mentors and mentees.**
(PDF)

## Acknowledgments

All authors contributed to the conceptualisation, direction and evaluation of the study. MW administered the study, collected the data and carried out the analysis. MW and NAA drafted the manuscript with review and editing contributions from all authors. We would like to thank all the mentors and mentees who have taken part in AIMMS Mentoring and in the evaluation. We would also like to acknowledge the strong support received from Founding Committee members' institutions, including their financial contributions to the running of the scheme and enabling staff to participate in the scheme and its creation. The views and opinions expressed in this review are those of the authors and do not necessarily reflect those of their institutions.

For the purpose of open access, the author has applied a CC BY public copyright licence to any Author Accepted Manuscript version arising from this submission.

## Author contributions

**Conceptualization:** Mirembe Woodrow, Elizabeth Benedikz, Louise D Bryant, Parbir Jagpal, Hannah Maria Jennings, Eleanor Monks, Vrinda Nayak, Musarrat Maisha Reza, Sikha Saha, Meena Upadhyaya, Kate Williams, John P Winpenny, Jane Illés, Reza Zamani, Nisreen A Alwan.

**Data curation:** Mirembe Woodrow.

**Formal analysis:** Mirembe Woodrow.

**Investigation:** Mirembe Woodrow, Nisreen A Alwan.

**Methodology:** Mirembe Woodrow, Nisreen A Alwan.

**Project administration:** Mirembe Woodrow, Nisreen A Alwan.

**Supervision:** Nisreen A Alwan.

**Writing – original draft:** Mirembe Woodrow, Nisreen A Alwan.

**Writing – review & editing:** Mirembe Woodrow, Elizabeth Benedikz, Louise D Bryant, Parbir Jagpal, Hannah Maria Jennings, Eleanor Monks, Vrinda Nayak, Musarrat Maisha Reza, Sikha Saha, Meena Upadhyaya, Kate Williams, John P Winpenny, Jane Illés, Reza Zamani, Nisreen A Alwan.

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
