## [Decision Letter · Decision Letter 0]

11 Jul 2024

PONE-D-24-01368‘I cannot be what I don’t see’: an evaluation of Academic Intersectionality Mentoring in Medical Schools (AIMMS Mentoring)PLOS ONE

Dear Dr. Alwan,

Thank you for submitting your manuscript to PLOS ONE. After careful consideration, we feel that it has merit but does not fully meet PLOS ONE’s publication criteria as it currently stands. Therefore, we invite you to submit a revised version of the manuscript that addresses the points raised during the review process.

We look forward to receiving your revised manuscript.

Kind regards,

Paavani Atluri

Academic Editor

PLOS ONE

Journal Requirements:

2. For studies involving third-party data, we encourage authors to share any data specific to their analyses that they can legally distribute. PLOS recognizes, however, that authors may be using third-party data they do not have the rights to share. When third-party data cannot be publicly shared, authors must provide all information necessary for interested researchers to apply to gain access to the data. (https://journals.plos.org/plosone/s/data-availability#loc-acceptable-data-access-restrictions)

3. Please amend the manuscript submission data (via Edit Submission) to include author John P Winpenny. 

Reviewers' comments:

Reviewer's Responses to Questions

**Comments to the Author**

1. Is the manuscript technically sound, and do the data support the conclusions?

Reviewer #1: Partly

Reviewer #2: Yes

Reviewer #3: Yes

Reviewer #4: Partly

2. Has the statistical analysis been performed appropriately and rigorously?

Reviewer #1: N/A

Reviewer #2: I Don't Know

Reviewer #3: Yes

Reviewer #4: N/A

3. Have the authors made all data underlying the findings in their manuscript fully available?

Reviewer #1: Yes

Reviewer #2: Yes

Reviewer #3: Yes

Reviewer #4: Yes

4. Is the manuscript presented in an intelligible fashion and written in standard English?

Reviewer #1: Yes

Reviewer #2: Yes

Reviewer #3: Yes

Reviewer #4: No

5. Review Comments to the Author

Reviewer #1: This is a very interesting study that describes a mentoring scheme developed for women from ethnic minority backgrounds. The study is interesting, but there are numerous points of improvement for publication. First, more needs to be done to describe the AIMMS and the evaluation of the scheme. The initial development and implementation of the scheme needs more attention.

Introduction

Line 60 – typo “ither” should be “other”

Line 69 – HE needs to be defined

Line 71 – HEI needs to be defined

Line 84 – Clarification is needed about the area that has limited published research. Is it regarding mentoring as a solution for addressing inequalities in the workplace? Or mentoring in the workplace in general?

Line 95 – NAA needs to be defined

Line 107 – From July 2021 - ?

The introduction needs a stronger flow in the first 3 paragraphs. The concept of intersectionality doesn’t span beyond the single paragraph, it seems disconnected from the rest of the intro despite seeming important to the scheme.

The description of the AIMMS should be all together. Everything that describes the overall mentoring program should not be split between the introduction and methods.

Methods

The scheme procedure section seems to bounce around between talking about how participants were recruited to be a part of the mentoring scheme and the evaluation.

How did women become aware of the scheme? How/who did they contact if they were interested? Is recruitment for the evaluation the same as recruitment for the scheme?

How was initial guidance provided? Email? Virtual or in-person meetings?

How long was the sign-up and matching process?

How were mentors identified?

Results

There are several universities identified as being a part of AIMMS were all of these institutions represented in the evaluation? Did they all have participants?

What information is being shared in the grey shaded box (lines 208-210).

What does sharing the age and ethnicity for the follow up questionnaire provide? Were different participants surveyed?

Unclear what Figures are presented in the text. There are titles for Figures 1, 2, and 3, but there are not any figures connected with the titles. There are grey shaded boxes spread throughout the results without labels/titles.

Lines 268-269 Were participants eligible to be both mentors and mentees?

What was the point in surveying participants prior to and after 6 months if no comparison of the data gathered is presented or discussed in the manuscript?

Discussion

There is no discussion of the information from the pre-survey. What did that information provide to the evaluation? Were mentees/mentors expectations met? Overall, more discussion of the results is needed.

Reviewer #2: Dear Authors,

I congratulate the authors for the wonderful initiative to promote gender equality in Medical and health sciences through mentoring aiming to highlight the significant hidden gaps in current environment in a developed country.

You conclude that mentoring between women from similar backgrounds is valuable and can assist with personal, professional and career development. I would agree with this statement but feel that this should change in the near future. Race, Gender or geographic background should be no longer an impediment but the should help develop a better scientific and global harmony among the scientific community leading to innovation, advancement and 'out of the box' thinking & mentoring advances.

This is a well designed and executed study to bring an important aspect of mentoring for women from ethnic minority backgrounds working in academic medicine and health sciences. You have acknowledged that this is a small number of 28 participants study with a scope for multi centric collaborative global initiative.

You explore various aspects of mentoring via formative and summative evaluation concluding that the AIMMS scheme promotes positive model for equitable environment and opportunity for women in health sciences.

This will open up multiple corridors for women empowerment and medical research.

We are not too far from this concept to become a global initiative breaking all the barriers.

I believe this study is a powerful step forward.

Can you make amendments:

1. Please add a paragraph before conclusion on Limitations of the study.

2. Please add a note on how this can be expanded to LMIC and across various medical and allied health faculties globally?

Thank you.

Best wishes.

Reviewer #3: visual representation of results shall be done. data illustration shall be improved. proper criteria for objective development shall be used. evaluation is not mentioned under the objective and aims section. what was the evaluation strategy, how was it measured?

Reviewer #4: Review Comments

The evaluation of the AIMMS Mentoring scheme provides valuable insights into the benefits of tailored mentoring for women from ethnic minority backgrounds in academic medicine and health sciences. While the study highlights significant positive outcomes, addressing the methodological limitations and expanding the scope of future research will enhance the understanding and impact of such initiatives.

With only 16 pairs (ten mentees and four mentors completing the follow-up), the sample size is relatively small, limiting the generalizability of the findings. The low number of participants who completed the follow-up questionnaire (14 out of 32) may introduce response bias, as those who did not respond could have different experiences. The study lacks Control Group: The absence of a control group makes it difficult to attribute observed outcomes solely to the mentoring scheme, as other factors might have influenced the results.

There is limited specificity: The findings are broadly positive but lack detailed statistical analysis or specific metrics, making it difficult to quantify the extent of development or satisfaction. The overwhelmingly positive feedback may overlook potential negative aspects or challenges that were not fully explored or reported. Providing more specific data and employing statistical methods would offer clearer insights into the scheme's effectiveness.

The study focuses on short-term outcomes, with limited insight into the long-term impact and sustainability of the mentoring relationships and their effects on career progression.

Conducting follow-up studies beyond six months would help assess the long-term impact and sustainability of the mentoring relationships.

6. PLOS authors have the option to publish the peer review history of their article (what does this mean? ). If published, this will include your full peer review and any attached files.

Reviewer #1: No

Reviewer #2: **Yes: ** Mr Chandrasekaran Kaliaperumal

Reviewer #3: No

Reviewer #4: No

---

## [Decision Letter · Decision Letter 1]

15 Jan 2025

‘I cannot be what I don’t see’: an evaluation of Academic Intersectionality Mentoring in Medical Schools (AIMMS Mentoring)

PONE-D-24-01368R1

Dear Dr. Alwan,

We’re pleased to inform you that your manuscript has been judged scientifically suitable for publication and will be formally accepted for publication once it meets all outstanding technical requirements.

Kind regards,

Claudia Noemi González Brambila, Ph.D.

Academic Editor

PLOS ONE

Additional Editor Comments (optional):

Reviewers' comments:

Reviewer's Responses to Questions

**Comments to the Author**

1. If the authors have adequately addressed your comments raised in a previous round of review and you feel that this manuscript is now acceptable for publication, you may indicate that here to bypass the “Comments to the Author” section, enter your conflict of interest statement in the “Confidential to Editor” section, and submit your "Accept" recommendation.

Reviewer #1: All comments have been addressed

Reviewer #3: All comments have been addressed

Reviewer #4: All comments have been addressed

2. Is the manuscript technically sound, and do the data support the conclusions?

Reviewer #1: Yes

Reviewer #3: Yes

Reviewer #4: Yes

3. Has the statistical analysis been performed appropriately and rigorously?

Reviewer #1: N/A

Reviewer #3: Yes

Reviewer #4: Yes

4. Have the authors made all data underlying the findings in their manuscript fully available?

Reviewer #1: Yes

Reviewer #3: Yes

Reviewer #4: Yes

5. Is the manuscript presented in an intelligible fashion and written in standard English?

Reviewer #1: Yes

Reviewer #3: Yes

Reviewer #4: Yes

6. Review Comments to the Author

Reviewer #1: The authors did a nice job revising their manuscript based upon reviewer feedback. No further edits are recommended.

Reviewer #3: We are now in a position to move forward with the acceptance of the publication, as the authors have diligently addressed all the critical points and ensured that every concern raised during the review process has been incorporated comprehensively. Over the course of this iterative process, the authors have demonstrated a commendable commitment to refining their work, taking into account the feedback provided at each stage.

Although the process has been extensive and has taken longer than initially anticipated, this additional time and effort have resulted in a robust and thoroughly detailed manuscript. The revisions reflect an in-depth understanding of the subject matter, meticulous attention to detail, and alignment with the expectations outlined by the reviewers.

This final version of the manuscript successfully integrates all the necessary components and highlights the authors’ capacity to present their research in a clear, cohesive, and impactful manner. The comprehensive nature of the revisions ensures that the publication is of high academic and professional quality, making it a valuable contribution to the field. As such, we are confident in proceeding with the acceptance of this work.

Reviewer #4: Authors have incorporated all the comments identified by the reviewer. The manuscript can be published

7. PLOS authors have the option to publish the peer review history of their article (what does this mean? ). If published, this will include your full peer review and any attached files.

---

## [Editor Report · Acceptance letter]

PONE-D-24-01368R1

PLOS ONE

Dear Dr. Alwan,

I'm pleased to inform you that your manuscript has been deemed suitable for publication in PLOS ONE. Congratulations! Your manuscript is now being handed over to our production team.

Kind regards,

on behalf of

Dr. Claudia Noemi González Brambila

Academic Editor

PLOS ONE